# Comprehensive Multi-Omic Evaluation of the Microbiota and Metabolites in the Colons of Diverse Swine Breeds

**DOI:** 10.3390/ani14081221

**Published:** 2024-04-18

**Authors:** Yanbin Zhu, Guangming Sun, Yangji Cidan, Bin Shi, Zhankun Tan, Jian Zhang, Wangdui Basang

**Affiliations:** 1Institute of Animal Husbandry and Veterinary Medicine, Tibet Academy of Agriculture and Animal Husbandry Science, Lhasa 850009, China; zhuyanbin126@126.com (Y.Z.); sungm1992@163.com (G.S.); 13889092363@163.com (Y.C.); 18989902737@163.com (B.S.); 2Faculty of Animal Science, Tibet Agricultural and Animal Husbandry University, Linzhi 860000, China; tanzhankun@xza.edu.cn; 3Animal Husbandry and Veterinary Station, Gongbujiangda, Linzhi 860000, China; zjian1971@126.com

**Keywords:** pigs, nutrition, microbiota, gut, SCFA

## Abstract

**Simple Summary:**

Pigs play a crucial role in human sustenance, with their intestinal microbiota composition significantly influencing their nutrition and metabolism. Recent research has focused on understanding these differences among various swine breeds to attain improved food quality and safety. We sought the microbial evidence of differential performances of Tibetan pigs. In this study, six Duroc × Landrace × Yorkshire (DLY) pigs and six Tibetan pigs at 200 days of age were examined. Analysis revealed distinct microbial compositions in DLY pigs, with higher levels of *Alloprevotella* and *Prevotellaceae_UCG-003*. This resulted in variations in short-chain fatty acids (SCFAs) concentrations, contributing to enhanced growth performance. Tibetan pigs exhibited increased abundance of *NK4A214_group*, leading to higher L-cysteine levels and the subsequent elevation of taurine in the colon and plasma. Taurine influences microbiota dynamics and metabolism, particularly in bile acid metabolism, showcasing Tibetan pigs’ proficiency in this area. Overall, DLY pigs excel in SCFA metabolism, while Tibetan pigs exhibit competence in bile acid metabolism. Leveraging these breed-specific differences could improve production performance in these pig breeds.

**Abstract:**

Pigs stand as a vital cornerstone in the realm of human sustenance, and the intricate composition of their intestinal microbiota wields a commanding influence over their nutritional and metabolic pathways. We employed multi-omic evaluations to identify microbial evidence associated with differential growth performance and metabolites, thereby offering theoretical support for the implementation of efficient farming practices for Tibetan pigs and establishing a robust foundation for enhancing pig growth and health. In this work, six Duroc × landrace × yorkshi (DLY) pigs and six Tibetan pigs were used for the experiment. Following humane euthanasia, a comprehensive analysis was undertaken to detect the presence of short-chain fatty acids (SCFAs), microbial populations, and metabolites within the colonic environment. Additionally, metabolites present within the plasma were also assessed. The outcomes of our analysis unveiled the key variables affecting the microbe changes causing the observed differences in production performance between these two distinct pig breeds. Specifically, noteworthy discrepancies were observed in the microbial compositions of DLY pigs, characterized by markedly higher levels of *Alloprevotella* and *Prevotellaceae_UCG-003* (*p* < 0.05). These disparities, in turn, resulted in significant variations in the concentrations of acetic acid, propionic acid, and the cumulative SCFAs (*p* < 0.05). Consequently, the DLY pigs exhibited enhanced growth performance and overall well-being, which could be ascribed to the distinct metabolite profiles they harbored. Conversely, Tibetan pigs exhibited a significantly elevated relative abundance of the *NK4A214_group*, which consequently led to a pronounced increase in the concentration of L-cysteine. This elevation in L-cysteine content had cascading effects, further manifesting higher levels of taurine within the colon and plasma. It is noteworthy that taurine has the potential to exert multifaceted impacts encompassing microbiota dynamics, protein and lipid metabolism, as well as bile acid metabolism, all of which collectively benefit the pigs. In light of this, Tibetan pigs showcased enhanced capabilities in bile acid metabolism. In summation, our findings suggest that DLY pigs excel in their proficiency in short-chain fatty acid metabolism, whereas Tibetan pigs exhibit a more pronounced competence in the realm of bile acid metabolism. These insights underscore the potential for future studies to leverage these breed-specific differences, thereby contributing to the amelioration of production performance within these two distinct pig breeds.

## 1. Introduction

The intestines of mammals harbor a substantial population of microorganisms, with reported microbial numbers reaching up to 10^13^–10^14^ [1]. As such, the intestinal ecosystem stands as one of the most extensive microbial habitats, playing an integral role in myriad physiological functions including digestion, absorption, and metabolic processes [2]. These microorganisms actively engage in interactions with the host, exerting a considerable influence over both disease susceptibility and normal physiological functions [3]. Furthermore, the establishment and diversity of these microbial communities are influenced by a variety of factors, including genetic lineage, age, gender, and feeding patterns. Furthermore, the gut microbiota serves as a producer of various metabolites, encompassing bile acids, fatty acids such as butyrate and other short chain fatty acids, and vitamins such as thiamine, folate, biotin, riboflavin, and pantothenic acid [4]. These compounds are generated through the co-metabolism of microorganisms and the host, ultimately impacting the overall health status of the host organism [5]. 

Metabolites encompass tangible compounds generated during the metabolic process, serving as both substrates and outcomes of metabolic reactions [6]. These compounds wield the capacity to instigate alterations in cellular functionalities, encompassing signal transduction, energy conversion, and cellular apoptosis [6]. In parallel, metabolomics stands as a discerning analytical methodology capable of holistically scrutinizing and contrasting end-stage products. This approach provides an intuitive depiction of the molecular metabolic profiles that arise from the metabolism of individuals, organ tissues, and cellular entities, thereby establishing a link between metabolic pathways and underlying biological mechanisms. Furthermore, it offers the ability to delineate an organism’s well-being by assessing the comprehensive metabolic network and conducting an exhaustive exploration of the organism’s metabolic processes [7].

Recent investigations have unveiled substantial disparities in the fecal microbial composition across distinct swine breeds [8,9,10]. Metabolites emerge as pivotal mediators that forge a connection between gut microorganisms and the physiological well-being of the host organism. Research has elucidated marked distinctions in both plasma and colonic metabolomes among diverse breeds of fattening pigs [11]. Furthermore, Yan et al. (2017) established significant discrepancies in microbiota in the gut makeup between Landrace and Meihua pigs, accompanied by notable differentials in microbial metabolite profiles [12]. Nevertheless, the specific differences between different pig breeds’ microbial communities and the metabolites that go along with them are yet unclear.

The commercial swine known as Duroc × Landrace × Yorkshire (DLY) pigs are distinguished by their excellent market retention rate. They exhibit advantages such as rapid growth, high lean meat percentage, and superior feed conversion efficiency [13,14]. On the other hand, the Tibetan pig is a local swine breed predominantly inhabiting the Qinghai–Tibet Plateau region of China. It demonstrates robust environmental adaptability and disease resistance. However, it is also burdened with disadvantages including low reproductive capacity and slow growth rate [15]. This study aimed to investigate the relationships between intestinal microbiota, metabolites, and plasma metabolic profiles in DLY pigs and Tibetan pigs, elucidating the differences in gut ecological niches and metabolic patterns. Additionally, it sought to unveil the microbial evidence associated with differential growth performance and metabolites between these two groups. The findings of this study offer a novel perspective for the swine industry to consider the Tibetan pig breed, thereby presenting prospects for the future scale-up of Tibetan pig farming, precision nutrition formulation, and improvement in husbandry practices.

## 2. Experimental Section

### 2.1. Animals

Twelve pigs, six of which were Tibetan and Duroc × Landrace × Yorkshire (DLY) pigs, were selected at random to be fed the same commercial diet when they were 200 days old (Zhengda, Tianjin, China). The animals were accommodated in pens furnished with plastic slatted flooring at the Tibet Liuya Agro-Pastoral Development Co., Ltd. (Shannan, China). The animals were randomly distributed into 2 experimental groups, and all animals were raised in the same house and fed the same diet, and were free to eat and drink during the experiment. The preliminary experiment lasted for 3 days and the formal experiment lasted for 5 weeks. No antibiotics were administered to the pigs throughout the 5-week experiment. The dietary composition adhered to the nutritional requirements outlined in the National Research Council (NRC) guidelines from 2012 (Table 1).

### 2.2. Sample Collection

At the final day, blood samples were collected via jugular venipuncture from each pig in an EDTA anticoagulation tube. After blood samples were centrifuged at 12,000 rpm for 10 min, they were quickly frozen in liquid nitrogen, and kept at −80 °C. Subsequently, samples of the colon digesta were taken out, quickly frozen in liquid nitrogen, and kept at −80 °C. The Tibet Academy of Agriculture and Animal Husbandry Science’s Institutional Animal Care and Use Committee (TAHS2021-15) approved all animal procedures, and all methods were performed in accordance with the relevant guidelines and regulations.

### 2.3. Untargeted Metabolomics Research Method and Parameter Setting

A 60 mg sample of colon digesta was meticulously weighed and subsequently transferred into a 1.5 mL Eppendorf tube. To the same tube, two diminutive steel balls were introduced. Following this, a solution of L-2-chlorophenylalanine (0.06 mg/mL) dissolved in methanol, serving as an internal standard, was added at 20 μL. Additionally, a composite mixture of methanol and water (4:1, *v*/*v*) totaling 650 μL, was incorporated into each individual sample. The samples were then subjected to cold storage at −20 °C for a duration of 2 min. Thereafter, they were subjected to grinding at 60 Hz for a span of 2 min. The subsequent step involved subjecting the entire set of samples to a 10 min extraction process using ultrasonication within an ice water bath, followed by further storage at −20 °C for a period of 5 h. Subsequently, the extracts were subjected to centrifugation at 4 °C (13,000 rpm) for a duration of 10 min. From each tube, supernatants amounting to 150 μL were meticulously collected using crystal syringes. These collected supernatants were then passed through sterile 0.22 μm microfilters and subsequently transferred into liquid chromatography vials. The vials, in turn, were stored at an ultra-low temperature of −80 °C until they were subjected to GC-MS analysis. 

Thawing was performed on plasma samples stored at −80 °C, allowing them to reach room temperature. A 150 μL aliquot of each sample was subsequently introduced into a 1.5 mL Eppendorf tube, wherein 10 μL of L-2-chlorophenylalanine (0.06 mg/mL) dissolved in methanol, serving as an internal standard, was meticulously added. The tube contents were then subjected to vortexing for a span of 10 s. Following this, a solvent mixture consisting of methanol and acetonitrile (2:1, *v*/*v*) totaling 450 μL was carefully introduced, and the resultant mixtures were subjected to vortexing for a duration of 1 min. Subsequently, the entire set of samples underwent a 10 min ultrasonic extraction within an ice water bath, followed by storage at −20 ℃ for a period of 30 min. The subsequent step encompassed centrifugation at 4 °C (13,000 rpm) for a duration of 10 min. From each glass vial, a 150 μL supernatant fraction was subjected to drying within a freeze concentration centrifugal dryer. Subsequently, a mixture of methanol and water (1:4, *v*/*v*) totaling 50 μL was introduced to each sample, followed by 30 s of vortexing. The samples were then subjected to a 3 min ultrasonic extraction within an ice water bath, followed by storage at −20 ℃ for a duration of 2 h. Thereafter, centrifugation was conducted at 4 °C (13,000 rpm) for a duration of 10 min. The resulting supernatants from each individual tube were meticulously collected through the utilization of crystal syringes. These collected supernatants were subsequently passed through 0.22 μm microfilters before being transferred into GC vials. The vials were subsequently stored at an ultra-low temperature of −80 °C in anticipation of GC-MS analysis [16,17].

### 2.4. Processing and Multivariate Analysis of Metabolomic Data

The initial GC-MS data underwent processing using Progenesis QI V2.3 software (Nonlinear Dynamics, Newcastle, UK) for tasks including baseline filtering, peak identification, integration, retention time correction, peak alignment, and normalization. Subsequently, following the description by Chawes (2019), parameters were established, and an analysis was conducted utilizing pertinent databases including the Human Metabolome Database (HMDB), Lipidmaps (V2.3), Metlin, EMDB, PMDB, and custom-built databases. This analytical process encompassed data processing, analysis, and comparative evaluation [16].

### 2.5. Detection of Short-Chain Fatty Acid Content in Samples

The extraction and quantification of SCFAs in the colonic chyme were conducted following established methodologies as outlined in a prior study [18]. Samples were extracted from approximately 0.5 g of colon content using ultrapure water. The resultant extract underwent centrifugation at 11,000× *g* rpm, followed by mixing with metaphosphoric acid (25%, *w*/*v*). A subsequent round of centrifugation at 12,500× *g* rpm was performed, and the resulting supernatant was filtered using a 0.45 µm Milled-LG filter (Millipore, Billerica, MA, USA). Subsequently, the analysis was carried out using an Agilent 7890 N gas chromatograph (Agilent, Santa Clara, CA, USA).

### 2.6. Quantitative Real-Time (qRT) PCR Analysis

Total RNA was extracted from the colon mucosa, using the RNeasy Mini Kit (GeneBetter, Beijing, China). The quantification of RNA sample concentrations was achieved using the NanoDrop 2000 spectrophotometer. Subsequently, cDNA synthesis was performed at 37 °C for 15 minutes, followed by a brief incubation at 85 °C for 5 seconds using the PrimeScriptTM RT reagent kit with gDNA Eraser (Thermo Fisher, Waltham, MA, USA). Detailed methods are outlined in Luo et.al (2021) [19]. The PCR primers used are shown in Appendix A.

### 2.7. Microbial 16S rRNA Gene Sequencing Analysis

Total genomic DNA was extracted using the manufacturer’s protocol with the EZNATM Soil DNA kit. For bacterial diversity analysis, The V3-V4 hypervariable regions of the bacterial 16S rDNA were amplified by a two-step PCR method using primers 338F (5′-ACTCCTRCGGGAGGCAGCAG-3′) and 806R (5′-GGACTACCVGGGTATCTAAT-3′) with unique 8-bp barcodes to facilitate multiplexing, and library sequencing and data processing were conducted by OE biotech Co., Ltd. (Shanghai, China). Sequences were analyzed and classified into operational taxonomic units (OTUs; 97% identity). Sequence data were analyzed with quantitative insights into microbial ecology (QIIME) package version 1.8.0 [20], using the Silva 138 reference database as a reference template [21]. The low abundant operational taxonomic units (OTU), identified by filtering OTU that had <10% of samples below 10 read counts, were removed. Tax4fun was used to predict functional profiles of microbial communities. Using BLAST, each representative read was annotated and tested against the Unite database (ITSs rDNA) (Blast 2015) (PRJNA872017).

### 2.8. Statistical Analysis

Data conforming to normal distribution were compared using Student *t*-test, while those with non-normal distribution were tested using the Kruskal–Wallis test. These analyses were performed using the JMP software (JMP R version 10.0.0, SAS Institute, Cary, NC, USA) for Windows. The correlation between the outcomes was examined using the psych package (Version 2.3.6) within the R environment (Version 4.3.1). The visual representation of the results was generated utilizing the pheatmap package (Version 1.0.12) within the same R environment (Version 4.3.1). All data were presented as mean ± standard error of the mean (SEM). Acceptable significance levels were at * *p* < 0.05, ** *p* < 0.01, and *** *p* < 0.001.

## 3. Results

### 3.1. Variations in Colonic Luminal Microbiome between DLY and Tibetan Pigs

Following size filtering, quality control, and chimera checking, 16S rRNA amplicon sequencing results revealed a total of 901,467 reads ranging from 73,049 to 77,051. Reads per sample were used to examine the species of pig breed in terms of microbial population in the colon. Sequencing counts were normalized to acquire normalized reads for each sample in operational taxonomic units (OTUs) based on 97% identity. As indicated in Figure 1, a Venn diagram was utilized to reveal the common and unique OTUs in two groups. In pigs from two groups, there were 2397 types of identical OTUs from over 3300 OTUs (Figure 1A). Regarding α-diversity, the microbiota diversity in DLY pigs was found to be notably higher when contrasted with that of Tibetan pigs (*p <* 0.05) (Figure 1B,C). In terms of β-diversity, there was a discernible differentiation in the microbiota composition between the two groups (Figure 1D,E).

At the phylum level, DLY pigs exhibited significantly elevated abundances of *Bacteroidetes*, *Fibrobacterota*, and *Campilobacterota*, coupled with considerably lower abundances of *Firmicutes* and *Deferribacterota*. Conversely, *Gemmatimonadota*, *Myxococcota*, and *Acidobacteriota* were exclusive to Tibetan pigs (Figure 1F,G). Among the top ten distinct microbial communities at the genus level, Tibetan pigs manifested reduced abundances in four microbial types: *p-251-o5*, *Prevotellaceae_UCG-003*, *Alloprevotella* and *Lachnospiraceae_NK4A136_group*. On the contrary, Tibetan pigs displayed augmented abundances in six microbial types: *UCG-002*, *Prevotellaceae_NK3B31_group*, *UCG-005*, *NK4A214_group*, *dgA-11_gut_group*, and *Streptococcus* (Figure 1H,I).

### 3.2. The Differential Metabolites in Colon between DLY and Tibetan Pigs

The disparities in colonic metabolites between the two pig breeds are depicted in Figure 2. The utilization of PCA score-plots and OPLS-DA highlighted distinct metabolomic profiles for both pig breeds (Figure 2A,B). Furthermore, a comparison between Tibetan and DLY pigs revealed 60 up-regulated and 10 down-regulated metabolites in Tibetan pigs (VIP ≥ 1, *p* value ≤ 0.05) (Figure 2C,D). The primary differential metabolites are presented in Appendix A. In comparison to DLY pigs, Tibetan pigs had considerably reduced relative levels of three carbohydrates (maltotriose, D-tagatose, and methyl beta-d-glucopyranoside), two amino acids (tranexamic acid and ornithine), and five additional substances. Meanwhile, in Tibetan pigs, the relative levels of 13 amino acids, two benzoic acids, one bile acid, three carbohydrates, 13 fatty acids, one organosulfonic acid, one cholestane steroids and 26 other compounds were significantly higher than those in DLY pigs.

These differential metabolites were related to various pathways based on KEGG analysis (Figure 2E), including differential metabolites mainly enriched in protein and bile acid metabolism.

### 3.3. Differential Plasma Metabolites between DLY and Tibetan Pigs

The distinctive plasma metabolomic profiles between the two pig breeds are illustrated in Figure 3. PCA score-plots and OPLS-DA show that both breeds of pigs had differential metabolites (Figure 3A,B). Compared with DLY pigs, Tibetan pigs had 39 up-regulated and 37 down-regulated metabolites (VIP ≥ 1, *p* value ≤ 0.05) (Figure 3C,D). Notably, the key differential metabolites are detailed in Appendix A. In Tibetan pigs, the relative levels of five amino acids, eight carbohydrates, two fatty acids, one organosulfonic acid, one cholestane steroid, and 22 other compounds were significantly higher compared to those in DLY pigs. Moreover, Tibetan pigs displayed significantly lower relative levels of six amino acids, eight carbohydrates, two fatty acids and 21 other compounds in comparison to DLY pigs. All these differential metabolites exhibited enrichment in diverse pathways, including purine, galactose, cysteine metabolism, and the TCA cycle, as indicated by KEGG analysis (Figure 3E). 

In the differential metabolites, only three metabolites showed the same trends (increased) in both colonic digesta and plasma, which were hexadecanedioic acid, linoleic acid, and taurine.

### 3.4. The same Metabolism Pathways in Both Plasma and Colon

There were six pathways of KEGG enrichment found in the analysis results of differential metabolites that were the same in both plasma and colonic contents, which were ABC transporters, glucagon signaling pathway, alanine, aspartate and glutamate metabolism, central carbon metabolism in cancer, citrate cycle (TCA cycle), and cysteine and methionine metabolism. In these six pathways, there were 13 metabolites in the colon (pyruvic acid, L-valine, L-aspartic acid, serine, taurine, 4-aminobutyric acid, L-methionine, 4-hydroxyproline, L-lactic acid, L-cysteine, phthalic acid, isocitric acid, 2-ketobutyric acid, and digalacturonic acid) and 12 metabolites in the plasma (glucose-1-phosphate, sulfuric acid, glycerol, fumaric acid, L-histidine, malic acid, N-carbamoylaspartate, L-asparagine, homocystine, taurine, L-cystine, and erythritol). Compared with the three common differential metabolites showing the same trends in both plasma and colon, only taurine was enriched in the pathways based on KEGG analysis.

### 3.5. Differential Concentrations of SCFAs in the Colonic Content of Two Different Species Pigs

Figure 4 illustrates the concentrations of SCFAs within the colons of the two pig groups. In comparison to Tibetan pigs, DLY pigs exhibited notably elevated concentrations of acetic acid (*p* < 0.05), propionic acid (*p* < 0.01), and total SCFAs (*p* < 0.05) (Figure 4A,B,D). While DLY pigs also demonstrated higher butyric acid concentrations, this difference did not reach statistical significance (*p* = 0.078) (Figure 4C). In the case of other SCFAs, no statistically significant differences were observed (*p* > 0.05); however, there were discernible trends indicating higher concentrations of isobutyric acid and hexanoic acid in DLY pigs compared to Tibetan pigs, as illustrated in Appendix A. The proportions of SCFA concentrations displayed similar patterns in both breeds. Acetic acid was present at 65.22% and 61.72%, propionic acid at 16.50% and 16.59%, and butyric acid at 10.58% and 11.39% in the DLY pigs, while other SCFAs’ concentrations were all lower than 5% (Figure 4E,F). 

### 3.6. Gene Expression Levels Related to Metabotropic Receptors in Mucosal Tissues

Figure 5 displays the gene expressions associated with metabotropic receptors. The expressions of the genes *TGR5* (*p* = 0.031) and *FXR5* (*p* = 0.0004) were much higher in Tibetan pigs (Figure 5A,B). On the other hand, DLY pigs exhibited significantly higher gene expressions of *GPR41* (*p* = 0.0039) and *GPR43* (*p* = 0.0065) (Figure 5C,D). In terms of *SLC5A8* expression, Tibetan pigs exhibited a higher level, although the difference was not statistically significant (Figure 5F). As for the expressions of other genes, no significant differences were observed (*p* > 0.05) (Figure 5E,G). 

### 3.7. Correlations between Microbial Communities, Colonic Metabolites, and Plasmatic Metabolites

The interactions between differentially abundant microbes, differential colonic metabolites, and differential plasmatic metabolites are shown in Figure 6. *Streptococcus*, *UCG−002*, *UCG-005* and *NK4A214_group* showed significantly high correlations with most metabolites in either the colon or plasma (Figure 6A,B). In particular, the four microbes were all significantly positively correlated with L-cysteine and taurine in the colon, in which *NK4A214_group* showed the highest correlation (Figure 6B). Furthermore, only *NK4A214_group* and *Streptococcus* showed significantly positive correlations with taurine in plasma (Figure 6A). Meanwhile, taurine in either plasma or colon showed positive correlations with amino acids (Figure 6C), but the correlation between taurine in two tissues was not significant; instead, they were all positively correlated with L-cysteine (Figure 6D). 

### 3.8. Correlations between Lithocholic Acid and Microbial Communities, Metabolites, or Receptor Related Genes

Then, we analyzed the relationships between lithocholic acid and colonic microbes (Figure 7A), colonic metabolites (Figure 7B), plasmatic metabolites (Figure 7C), and related genes (Figure 7D). As shown in Figure 7, we found that lithocholic acid did not have a significant correlation with taurine in the colon (Figure 7C), but it was significantly positively correlated with *NK4A214_group* (Figure 7A), taurine in plasma (Figure 7B), and L-cysteine in colon (Figure 7C). Finally, it had highly significantly positive correlations with *FXR5* and *TGR5* (Figure 7D).

### 3.9. Correlations between Short Chain Fatty Acids, Microbial Communities, and Receptor-Related Genes

Figure 8 shows the associations between SCFAs, colonic microbes and receptor genes. For significantly different concentrations of SCFAs, which were acetic acid, propionic acid and total SCFAs, they were all significantly positively correlated with three less abundant microbes in Tibetan pigs (Prevotellaceae_UCG-003, Alloprevotella, and Lachnospiraceae_NK4A136_group) (Figure 8A) and two related genes (GPR41 and GPR43) (Figure 8B), in which Prevotellaceae_UCG-003 showed the highest correlations, and it also had positive correlations with the other three SCFAs (isobutyric acid, butyric acid, and isovaleric acid). However, only Prevotellaceae_UCG-003 and Alloprevotella showed significantly positive correlations with GPR41 and GPR43 (Figure 8C). Meanwhile, SCFAs (acetic acid, propionic acid and total SCFAs) were significantly negatively correlated with four more abundant microbes in Tibetan pigs (UCG−002, NK4A214_group, dgA−11_gut_group, and Streptococcus) (Figure 8A), and except for dgA−11_gut_group, the correlations between the other three more abundant microbes and related genes (GPR41 and GPR43) were significantly negative (Figure 8C).

## 4. Discussion

In our study, we aimed to explore the differences in gut microbiome composition and metabolites among pigs of different genetic backgrounds and their implications for host physiology. Previous research has underscored variations among pig breeds [22,23,24]. Building on this foundation, our findings demonstrate significant disparities in metabolites, short-chain fatty acids (SCFAs), and receptor expression levels in both colonic and plasma environments across distinct pig breeds. Notably, these differences correlate with divergent production performances observed in these breeds. 

Numerous studies have shown that intestinal function is positively correlated with gut microbial abundance, and that higher gut microbial diversity and richness lead to better nutrient absorption [25,26]. In essence, microbes can be regarded as an additional organ within the host organism. Noteworthy parallels can be drawn between our findings and previous research, which has identified divergent microbial profiles in different pig breeds [12,24]. Our experimental outcomes further underscore these similarities. Specifically, our investigation revealed notable discrepancies in colonic microbial compositions, notably a diminished α-diversity index in Tibetan pigs. Among the six microbial strains exhibiting higher abundances in Tibetan pigs, *UCG-002* and *UCG-005* have been linked to cholesterol levels [27]; a finding consonant with our observation of elevated cholesterol levels in Tibetan pigs. Elevated cholesterol concentrations are associated with several diseases, including diabetes and cardiovascular disorders [28]. Concurrently, the presence of *Lachnospiraceae_NK4A136_group* has been associated with the fortification of the host’s intestinal barrier [29]. This observation implies that DLY pigs may possess superior gut health in comparison to Tibetan pigs.

The variances observed in colon and plasma metabolites can be attributed to multiple factors, including microbes, dietary factors, and age, collectively impacting various physiological functions. In this study, the two groups of experimental animals were selected at similar ages and fed the same commercial diet. Our findings suggest that certain alterations may be linked to microbial influences, with the resultant metabolic shifts offering insights into the disparities between the two breeds. Notably, elevated levels of specific amino acids, particularly essential amino acids such as methionine and threonine, may facilitate enhanced efficiency in utilizing other amino acids among pigs [30]. In our research, the KEGG enrichment result shows that different metabolites were enriched in cysteine and methionine metabolism and alanine, aspartate and glutamate metabolism in either the plasma or colon, and the concentrations of most enriched metabolites were higher in Tibetan pigs in either plasma (four higher level vs. two lower level) or colon (seven higher level vs. zero lower level). Especially in the colons of Tibetan pigs, the concentrations of all enriched metabolites were higher, which indicates that Tibetan pigs have a better protein metabolism capacity. Moreover, for the energy and carbohydrate metabolism, ABC transporters, Glucagon signaling pathway, and TCA cycle were enriched in both plasma and colon KEGG enrichment results. Most metabolites enriched in these three pathways were increased in Tibetan pigs (six higher level vs. three lower level in plasma, nine higher level vs. zero lower level in colon). Therefore, Tibetan pigs have a better energy metabolism capacity. Zeng et al. (2020) also found that energy metabolism, amino acid metabolism, and carbohydrate metabolism were consistently enriched at high altitudes in both Tibetans and Tibetan pigs. High altitude will result in unique gut bacteriomes and functions [31]. Therefore, these two better capacities may be related to the fact that the microbes are affected by high altitude. Afterwards, the colons of Tibetan pigs had two different types of higher-level benzoic acids and derivatives, which reduced the microbiota’s variety [32], which might explain the result whereby Tibetan pigs had lower α-diversity in the colon. Furthermore, two kinds of fatty acids were increased in both plasma and colon (hexadecanedioic acid and linoleic acid), which supply energy to animals. Nevertheless, excessive fatty acid concentrations cause a number of illnesses and increase fat tissue, which explains why Tibetan pigs have higher adipose tissue [33].

In Tibetan pigs, taurine was increased in both the plasma and colon, which indicates that taurine is a key metabolite supporting the differences between the two breeds of pigs. Firstly, *UCG-005* was significantly positively correlated with taurine in the colon. Therefore, it can enhance the concentration of taurine in the colon. However, the correlations between taurine in plasma and microbes or taurine in colon were weak based on the correlation analysis; on the other hand,, L-cysteine in the colon was significantly positively correlated with plasma (r = 0.82, *p* = 0.007) and colon (r = 0.74, *p* = 0.007), respectively, which indicates that taurine in the colon does not directly affect the concentration of taurine in plasma, and L-cysteine is the main factor affecting the concentrations of taurine in two tissues. This is because L-cysteine is the upstream metabolite of taurine, which can enhance the production of taurine [34]. Then, we analyzed the correlations between L-cysteine and microbes; we found that four microbes (*Streptococcus*, *UCG−002*, *UCG-005* and *NK4A214_group*) were significantly positively correlated with L-cysteine, especially *NK4A214_group*, which showed the highest correlation. Therefore, although microbes can affect the concentration of taurine in colon, the main factor affecting taurine metabolism in both the colon and plasma is L-cysteine, which is enhanced by *NK4A214_group*. Furthermore, taurine was found to increase the abundance of microbes in the colon, such as *NK4A214_group* and *Ruminococcus* [35]. Therefore, taurine conversely maintains microbial homeostasis. Finally, taurine was found to maintain the protein metabolism, which is similar to our result that taurine showed positive correlations with amino acids [36]. Therefore, Tibetan pigs with a higher level of taurine can support better protein metabolism to maintain growth and health.

Elevated taurine levels may improve the bile acid metabolic system, which would help pigs by protecting the body by providing microbes energy and warding off pathogens [37]. In our result, Tibetan pigs also had higher level of lithocholic acid in the colon. Therefore, we further analyzed the interactions between lithocholic and microbes or metabolites. We found that in only the *NK4A214_group*, taurine in plasma and L-cysteine in colon showed significant correlations with lithocholic acid, which indicates that taurine in the plasma, and not in the colon, directly affects the concentration of lithocholic acid. Furthermore, lithocholic acid can protect intestinal mucosa by inhibiting epithelial apoptosis [38]. Additionally, by sensitizing BE(2)-m17 and SK-n-MCIXC cells to hydrogen peroxide, which may cause cell death, lithocholic acid can kill gliomas without endangering normal neurocytes. Normal neuronal cells are resistant to this kind of cell death [39,40]. Therefore, lithocholic acid can support Tibetan pigs in maintaining health. Furthermore, taurine can affect lipid metabolism by conjugating bile acids [41], which proves that Tibetan pigs have more adipose tissue. Finally, to prove the increase in lithocholic acid in Tibetan pigs, we analyzed two related genes’ expressions (*TGR5* and *FXR5*). These two genes were expressed significantly more highly in Tibetan pigs. *FXR* is a nuclear receptor highly expressed in the liver and the intestine [42]. *TGR5* and *FXR5*, which maintain glucose homeostasis, are impacted by bile acid [43]. Specifically, bile acid stimulates FXR to suppress proglucagon transcription and *GLP-1* synthesis, and stimulates TGR5 to increase *GLP-1* secretion [44]. Thus, the increased lithocholic acid in Tibetan pigs can be explained by the higher level of mRNA expression of *TGR5* and *FXR5*. In our correlation result, lithocholic acid also showed highly significantly positive correlations with these two genes.

The pivotal role of short-chain fatty acids (SCFAs) as key mediators connecting disease, nutritional aspects, and gut microbiota hinges on their concentrations, which are highly influenced by multifactorial determinants, including microbes, age, and dietary constituents. Our findings highlight microbes as the primary determinants shaping SCFA concentrations. In particular, diminished levels of SCFA-producing microbes within the colons of Tibetan pigs contribute to lower SCFA concentrations, corroborating findings from prior research [15]. Fundamentally, elevated levels of total SCFAs within the colons of DLY pigs are indicative of enhanced growth performance. [45]. More specifically, the decreased contents of acetic acid and propionic acid in Tibetan pigs correlate with compromised growth and health, aligning with other studies [46]. Moreover, the presence of butyric acid, an SCFA, not only provides energy for the host, but also bolsters host defense by promoting the production of defense peptides [47]. Therefore, higher concentrations of butyric acid could support DLY pigs’ health. Moreover, many of the regulatory properties of SCFAs require signaling through *GPRs*, including *GPR43*, *GPR41*, and *GPR109* [48]. These receptors are expressed across various cell types, including the intestinal epithelium and immune cells. Their interaction with GPRs fortifies the intestinal epithelial barrier and upholds intestinal equilibrium [49]. In Tibetan pigs, the decreasing expressions of *GPR41* and *GPR43* can explain the lower concentrations of SCFAs, and our correlation results also show that there were significantly positive correlations with SCFAs (acetic acid, propionic acid, butyric acid, and total SCFAs) and related genes (*GPR41* and *GPR43*).

To further identify correlations between microbes and SCFAs metabolism, we analyzed the relations between microbes and SCFAs or related genes. We found that, in Tibetan pigs, three more abundant microbes (*UCG−002*, *NK4A214_group*, and *Streptococcus*), especially *Streptococcus,* were significantly negatively correlated with SCFAs (acetic acid, propionic acid and total SCFAs) and related genes (*GPR41* and *GPR43*), which indicates that SCFAs may affect the abundances of some of these three microbes. Another study also found that SCFAs can inhibit the biofilm formation of *Streptococcus gordonii,* which supports our results [50]. Meanwhile, higher abundances of these microbes may lead to lower abundances of SCFA producers, thus reducing SCFAs. In our results, the abundances of two other SCFA producers were decreased in Tibetan pigs, which were *Prevotellaceae_UCG-003* and *Alloprevotella*. All of them were significantly positively correlated with SCFAs (acetic acid, propionic acid and total SCFAs) and related genes (*GPR41* and *GPR43*), which indicates that these two microbes can enhance the production of SCFAs by stimulating the expression of *GPR41* and GPR*43*. Other researchers also attained similar results, finding that *Alloprevotella* and *Prevotellaceae_UCG-003* can produce SCFAs or affect SCFAs production [51,52]. While our study has provided valuable insights, it is not without limitations. A larger sample size of experimental animals would enhance the generalizability of our findings. Additionally, this study only compared Tibetan pigs with the common commercial breed DYL for reference. In consequence, in subsequent research, we will expand the sample size of experimental pigs based on the results of this study, delve deeper into the nutritional requirements of Tibetan pigs, and conduct comparative studies involving more breeds of pigs.

## 5. Conclusions

In our experiment, we found that both breeds of pigs had differential microbe, metabolite, SCFA and related gene expression levels. In DLY pigs, higher relative abundances of microbes containing Alloprevotella and Prevotellaceae_UCG-003 resulted in higher concentrations of SCFAs in acetic acid and propionic acid, as well as of total SCFAs, thus benefiting pigs. In Tibetan pigs, higher relative abundances of the NK4A214_group resulted in higher levels of L-cysteine, which enhance the production of taurine, and higher levels of taurine enhance the gut microbiota, protein metabolism, and lipid metabolism of pigs. Meanwhile, lithocholic acid, which is affected by taurines, also maintains pig health. Therefore, it is possible to use these differences to benefit either DLY pigs or Tibetan pigs, such as by using fecal microbiota transplantation to alter Tibetan pigs’ microbes and thus improve their growth performance. This study not only elucidates the metabolic differences in intestinal microorganisms between two pig breeds in the plateau region, but also provides a theoretical basis for future research on local pig breeds.

## Figures and Tables

**Figure 1 animals-14-01221-f001:**
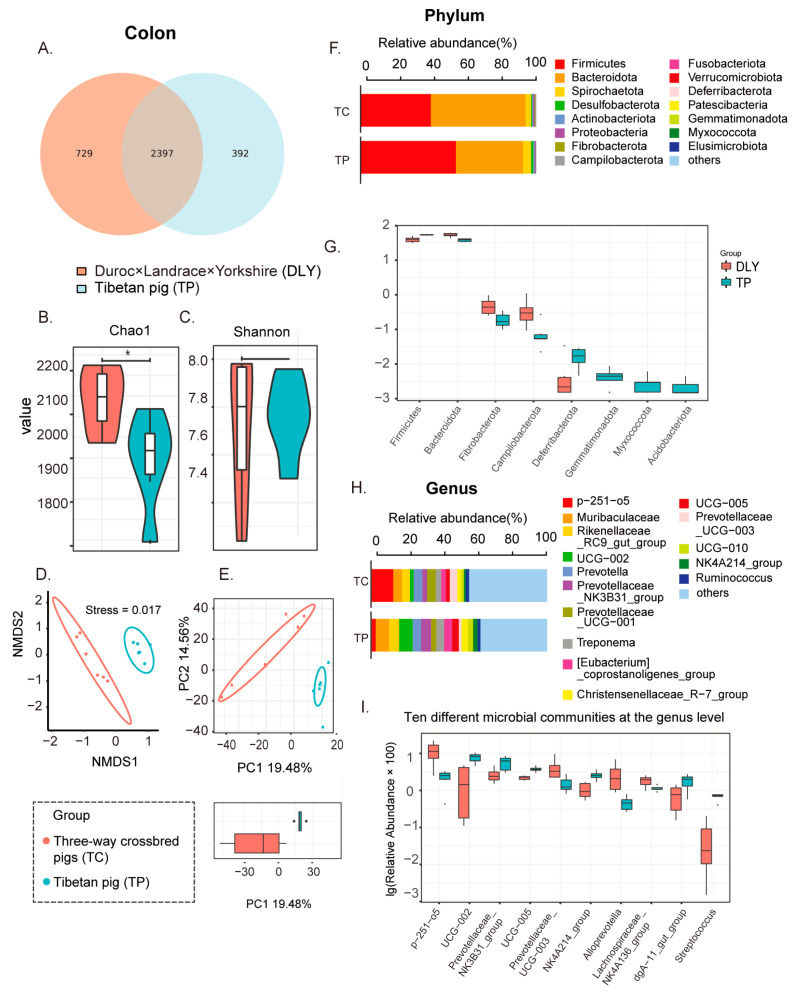
Differential microbiome in colon of two breeds of pigs. (**A**) Venn diagrams of two groups. (**B**) Chao1 index of two groups. (**C**) Shannon index of two groups. (**D**) NMDS results of two groups. (**E**) PCoA results of two groups. (**F**) Community analysis of two groups at the phylum level. (**G**) Top 10 differential microbes of two groups at phylum level. (**H**) Community analysis of two groups at genus level. (**I**) Top 10 differential microbes of two groups at genus level. Data are presented as mean ± SD and statistical significance was determined by the Wilcoxon rank-sum test; DLY pigs; TP, Tibetan pig; * *p* ≤ 0.05; *n* = 6.

**Figure 2 animals-14-01221-f002:**
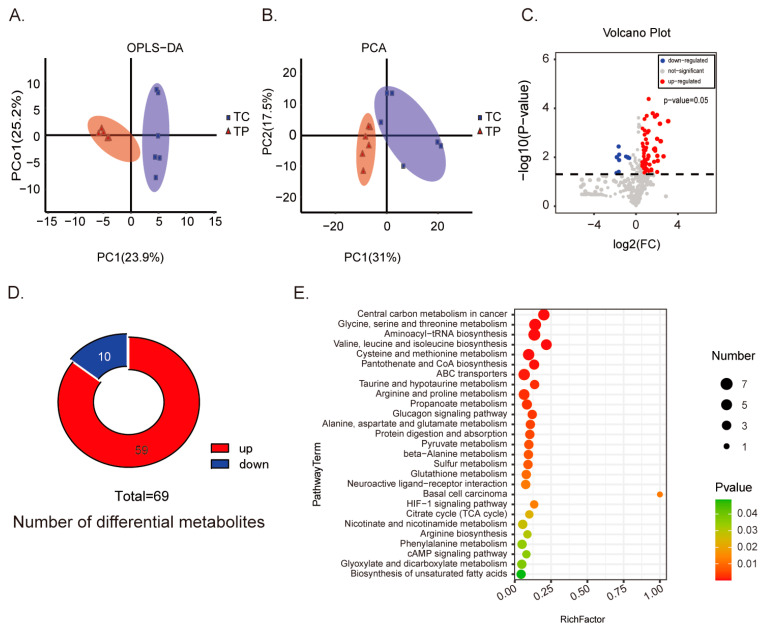
Differential metabolites in colon of two breeds of pigs. (**A**) OPLS-DA results of two groups. (**B**) PCA results of two groups. (**C**) Volcano plot of two groups. (**D**) Number of differential metabolites of two groups. (**E**) Top 27 possible pathways of metabolites of two groups. DLY pigs; TP, Tibetan pig; *n* = 6.

**Figure 3 animals-14-01221-f003:**
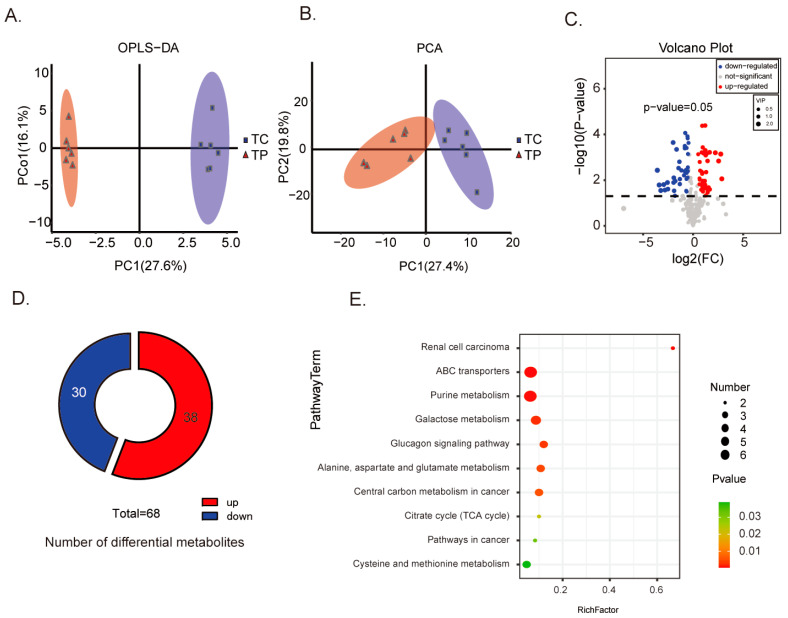
Differential metabolites in plasma of two breeds of pigs. (**A**) OPLS-DA results of two groups. (**B**) PCA results of two groups. (**C**) Volcano plot of two groups. (**D**) Number of differential metabolites of two groups. (**E**) Top 10 possible pathways of metabolites of two groups. DLY pigs; TP, Tibetan pig; *n* = 6.

**Figure 4 animals-14-01221-f004:**
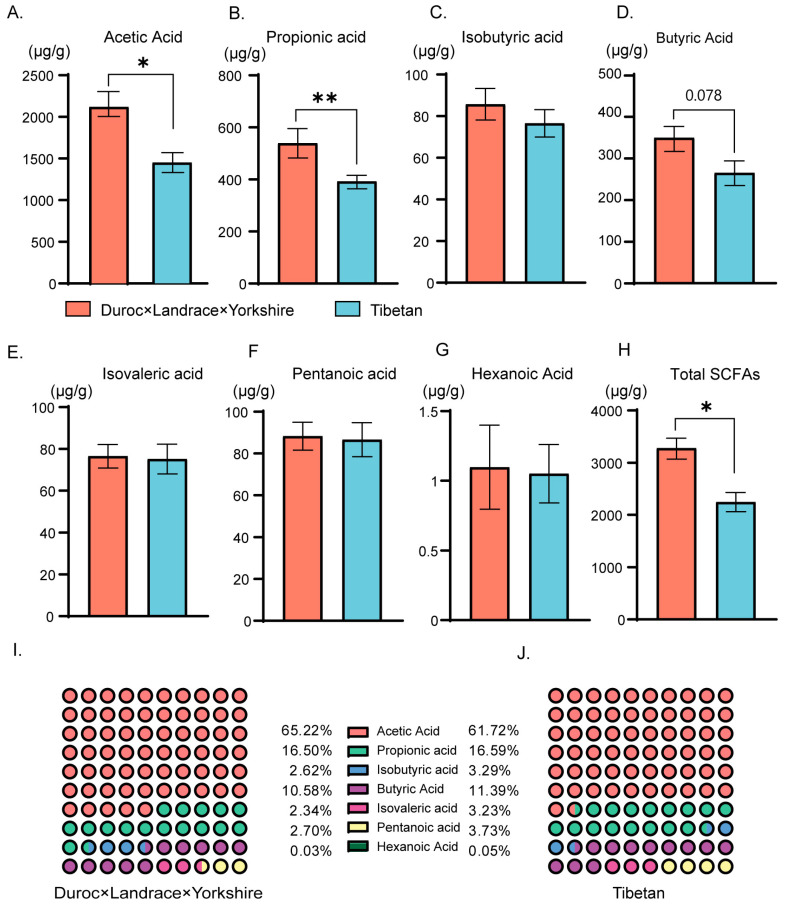
Differential concentrations of SCFAs in colons of two breeds of pigs. (**A**) Acetic acid (ug/g), (**B**) propionic acid (ug/g), (**C**) Isobutyric acid (ug/g), (**D**) Butyric acid (ug/g), (**E**) Isovaleric acid (ug/g), (**F**) Pentanoic acid (ug/g), (**G**) Hexanoic acid (ug/g), (**H**) total SCFAs (ug/g) in the colonic samples from DLY and TP pigs. (**I**) Percentages of SCFAs in DLY pigs. (**J**) Percentages of SCFAs in Tibetan pigs. Data are presented as mean ± SD and statistical significance was determined by the Wilcoxon rank-sum test; DLY pigs; TP, Tibetan pig; * represents significant difference (* *p* ≤ 0.05), ** *p* ≤ 0.01, *n* = 6.

**Figure 5 animals-14-01221-f005:**
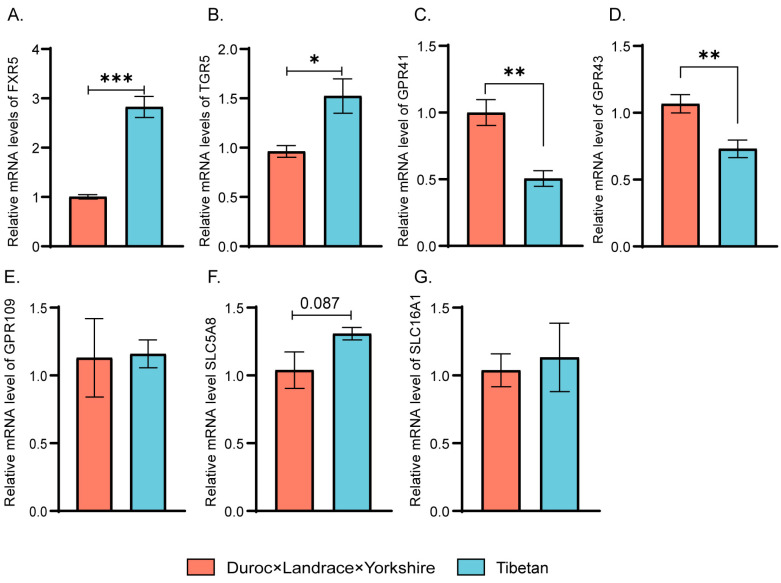
Differential expressions of related genes in mucosa of two breeds pigs. (**A**) mRNA expression of *FXR5*. (**B**) mRNA expression of *TGR5*. (**C**) mRNA expression of *GPR41*. (**D**) mRNA expression of *GPR43*. (**E**) mRNA expression *GPR109*. (**F**) mRNA expression *SLC5A8*. (**G**) mRNA expression *SLC16A1*. Data are presented as mean ± SD and statistical significance was determined by the Wilcoxon rank-sum test; DLY pigs; TP, Tibetan pig; * represents significant difference (* *p* ≤ 0.05), ** *p* ≤ 0.01 and *** *p* ≤ 0.001; *n* = 6.

**Figure 6 animals-14-01221-f006:**
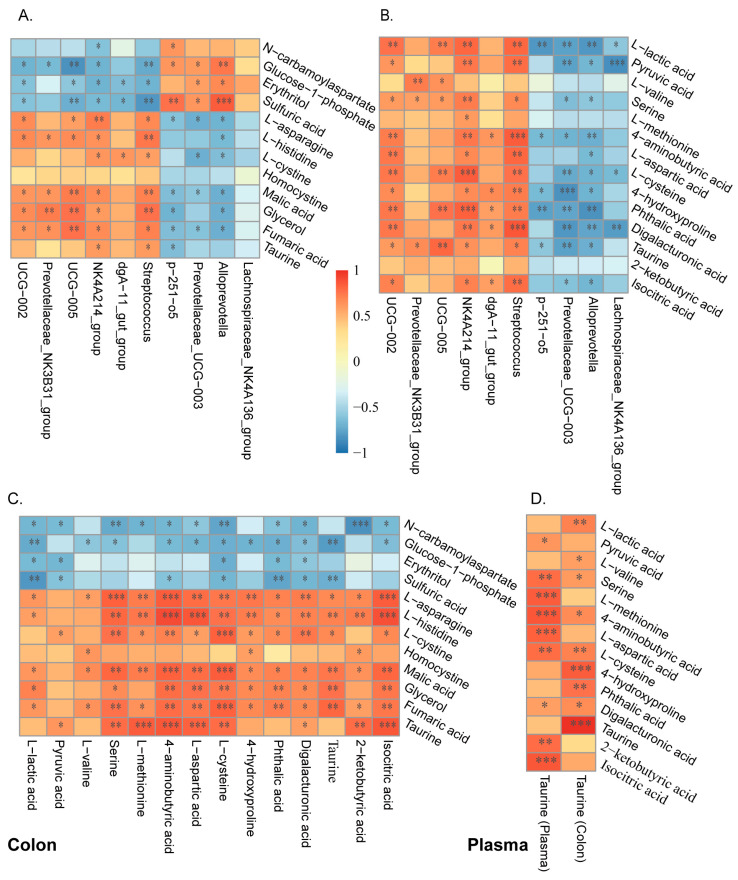
Correlation analysis of microbial communities, colonic metabolites, and plasmatic metabolites. (**A**) Interactions between colonic microbes and plasmatic metabolites. (**B**) Interactions between colonic microbes and colonic metabolites. (**C**) Interactions between colonic metabolites and plasmatic metabolites. (**D**) Interactions between colonic or plasmatic taurine and plasmatic metabolites. Correlations were determined by the spearman test, blue represents a negative correlation, red represents a positive correlation, *n* = 6. * *p* < 0.05, ** *p* < 0.01, and *** *p* < 0.001.

**Figure 7 animals-14-01221-f007:**
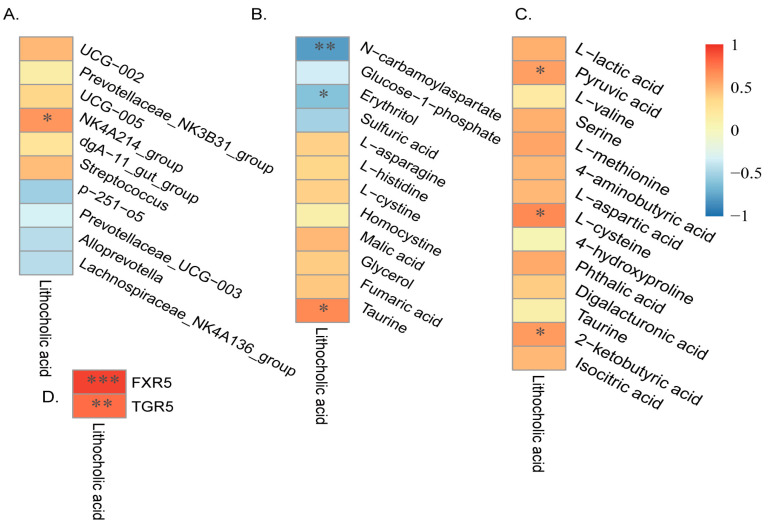
Correlations between lithocholic acid and microbial communities, metabolites, or receptor related genes. (**A**) Interactions between lithocholic acid and colonic microbes. (**B**) Interactions between lithocholic acid and plasmatic metabolites. (**C**) Interactions between lithocholic acid and colonic metabolites. (**D**) Interactions between lithocholic acid and related genes. Correlations were determined by the spearman test, blue represents a negative correlation, red represents a positive correlation, *n* = 6. * *p* < 0.05, ** *p* < 0.01, and *** *p* < 0.001.

**Figure 8 animals-14-01221-f008:**
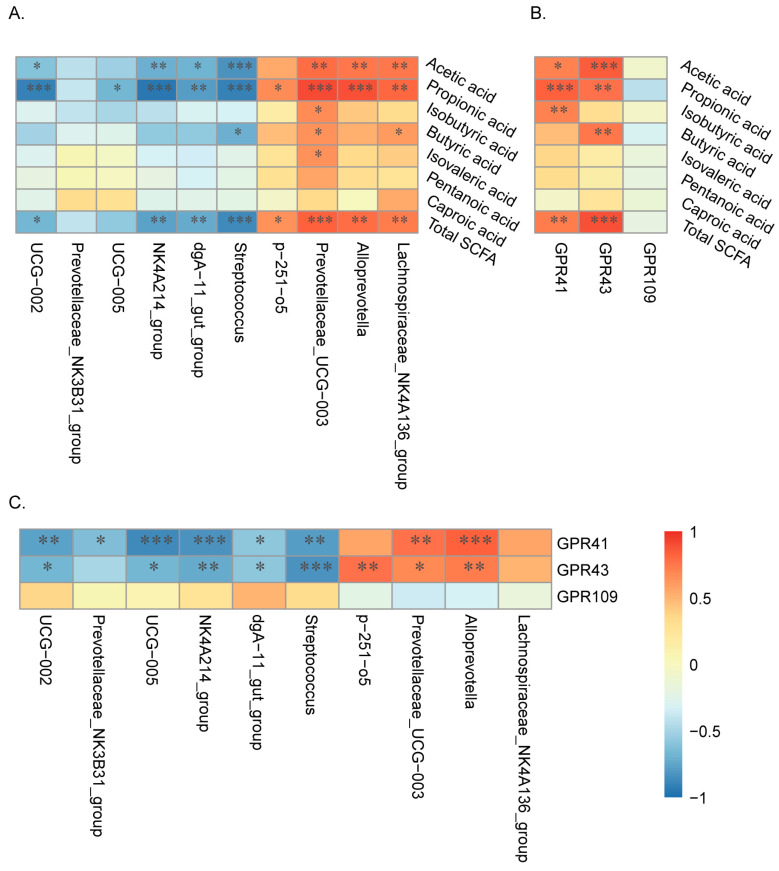
Correlations between short-chain fatty acids, microbial communities, and receptor-related genes. (**A**) Interactions between colonic microbes and SCFAs. (**B**) Interactions between SCFAs and related genes. (**C**) Interactions between colonic microbes and related genes. Correlations were determined by the spearman test, blue represents a negative correlation, red represents a positive correlation, *n* = 6. * *p* < 0.05, ** *p* < 0.01, and *** *p* < 0.001.

**Table 1 animals-14-01221-t001:** Composition and nutrient levels of basal diets (basis).

Item	Content
Corn	74.00
Soybean meal	15.00
Wheat bran	7.0
Premix	4.0
Total	100.0
Nutrient levels	
DE/(MJ/kg)	14.78
Dry matter (%) DM	87.68
Crude protein (%)	15.11
Ca	0.53
TP	0.42
L-lysine	0.72
Methionine	0.24
Threonine	0.53
Tryptophan	0.05

The premix provided the following per kilogram of the diet: vitamin A. 3000 IU; vitamin D, 300 IU; vitamin E, 38.5 mg; vitamin K3, 1.35 mg; vitamin B1, 2.5 mg; vitamin B2, 6.5 mg; vitamin B6, 3.0 mg; vitamin B12, 0.025 mg; nicotinic acid, 25 mg; pantothenic acid, 15 mg; biotin, 0.75 mg; L-lysine monohydrochloride, 0.68 g; Cu (as copper sulfate), 5 mg; Fe (as ferrous sulfate), 100 mg; Zn (as zinc sulfate), 80 mg; Mn (as manganese sulfate), 20 mg; I (as potassium iodide), 0.5 mg; Se (as sodium selenite), 0.3 mg. DM, CP, Ca, and TP were measured values, while the others were calculated values.

## Data Availability

The data presented in this study are available on request from the corresponding author. The data are not publicly available due to specific ethical and privacy considerations.

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
