# Peer review of "Comprehensive Multi-Omic Evaluation of the Microbiota and Metabolites in the Colons of Diverse Swine Breeds"

_animals, 2024, doi:10.3390/ani14081221_

Round 1
Reviewer 1 Report
Comments and Suggestions for Authors
In the methodology, the variety of techniques used to detect the microbiome, metabolites and genes is appropriate.
The results are presented with excessive graphics.
The discussion seems to me to be well presented in its comparison with similar studies.
Finally, I consider that the number of animals used in the study is insufficient to obtain general conclusions about the differences between breeds of pigs.
Author Response
- In the methodology, the variety of techniques used to detect the microbiome, metabolites and genes is appropriate.
Response: Thanks for this comment. We appreciate the affirmation.
- The results are presented with excessive graphics.
Response: Thanks for the advice, we tried to reduce some graphics contain Figure 2C, Figure 2D, Figure 2H, Figure 3C, Figure 3D, and Figure 3H, and the related figure legends and descriptions in line “” and line “” has been changed.
- The discussion seems to me to be well presented in its comparison with similar studies.
Response: Thanks for this comment. We appreciate the affirmation.
- Finally, I consider that the number of animals used in the study is insufficient to obtain general conclusions about the differences between breeds of pigs.
Response: Thank you very much for the valuable suggestions provided by the reviewer. Indeed, studies involving six animals per experimental group also exist in the literature (1. DOI: 10.3109/19401736.2014.913169 2. DOI: 10.1111/jpn.13283 3. DOI: 10.1186/s40104-021-00581-3). Therefore, we believe that having six pigs per group in our study is appropriate. However, we are willing to adopt the editor's suggestion for increasing the number of animals in each experimental group in future experiments to enhance the persuasiveness of our research. We appreciate the constructive feedback and are willing to make adjustments accordingly in our future research.

Reviewer 2 Report
Comments and Suggestions for Authors
In research authors focuses on comparing microbiota and metabolites in certain two pig breeds. However, in Introduction is lack of precise aim of the study. Instead of it, a chapter describing hypothetic (line 96-104) advantages of microbiota profile knowledge. As authors mentioned in Discussion (line 415), previous studies reveled divergent microbial profiles in different pig breeds. Therefore, the purpose of the study becomes unclear, and it is difficult to evaluate the originality/ novelty of research.
The significant of content in this research is quite low, as long as the main aim of research is comparing the microbiome in breeds, however, it could be a good base for further research concerning f.e. the influence of microbiota on productive performance in certain breeds etc.
The quality of presentation is on high level, brief and clear chapters, interesting figures improve the quality of paper, and made the article interesting and understandable for reader.
As long as the conclusions are not clear and focuses only on divergencies in microbial and metabolic profile in certain breeds, the overall merit of work in quite low and unsatisfactory. In Discussion section the authors highlighted the differences in metabolites in colon and plasma but indicated that those results may be influenced by multiple factors (line 427), thus, unfortunately, the conclusions sound rather like hypothesis then like facts.
I would recommend rearrange the research, because, in did, investigating the microbial profile and it influence on animal health, productive performance, public health management is quite interesting topic. But, unfortunately, the article in present form cannot be accepted.
Detailed comments:
Line 40 please change the font size
Line 95 please delete point before bracket with citation
Line 408 In reviewer opinion this conclusion sounds rather like hypothesis, as long as the main of study was to assess the microbiota of different breeds, not the influence of modified microbiota on performance of growth etc. The same comment goes to line 32-33 in abstract, however, that could be the further/ next step of investigation, and it would be novel and interesting.
Author Response
- In research authors focuses on comparing microbiota and metabolites in certain two pig breeds. However, in Introduction is lack of precise aim of the study. Instead of it, a chapter describing hypothetic (line 96-104) advantages of microbiota profile knowledge. As authors mentioned in Discussion (line 415), previous studies reveled divergent microbial profiles in different pig breeds. Therefore, the purpose of the study becomes unclear, and it is difficult to evaluate the originality/ novelty of research.
Response: Thanks for this comment. We have rewritten this section to make our points of innovation and the focus of this article clearer and more prominent (line 96-103).
- The significant of content in this research is quite low, as long as the main aim of research is comparing the microbiome in breeds, however, it could be a good base for further research concerning f.e. the influence of microbiota on productive performance in certain breeds etc.
Response: Thanks for this comment. Indeed, our research group will delve deeper into the study of production performance and digestive characteristics of Tibetan pigs in future investigations, analyzing the nutritional requirements of Tibetan pigs at different stages.
- The quality of presentation is on high level, brief and clear chapters, interesting figures improve the quality of paper, and made the article interesting and understandable for reader.
Response: Thank you for your feedback. Yes, this experiment serves as the foundation for our ongoing work, and we are committed to pressing forward.
- As long as the conclusions are not clear and focuses only on divergencies in microbial and metabolic profile in certain breeds, the overall merit of work in quite low and unsatisfactory. In Discussion section the authors highlighted the differences in metabolites in colon and plasma but indicated that those results may be influenced by multiple factors (line 427), thus, unfortunately, the conclusions sound rather like hypothesis then like facts.
Response: Thanks for your comments, what we focused on is finding microbial evidence leading to differential growth performance or health of Tibetan pigs. Of course there should be other factors affecting the pigs’ growth. However, we only want to make sure we can link microbes with other parameters trying to use microbes explain other differential parameters.
- I would recommend rearrange the research, because, in did, investigating the microbial profile and it influence on animal health, productive performance, public health management is quite interesting topic. But, unfortunately, the article in present form cannot be accepted.
Response: We appreciate the reviewer's insightful suggestion. However, we respectfully disagree with the recommendation to rearrange our research focus. Firstly, it's worth noting that Tibetan pigs and Duroc pigs have significantly different growth cycles and adult weights, rendering direct comparisons of growth performance data less meaningful. Secondly, our decision to select animals of similar ages allows for a more robust comparison of microbial profiles and metabolic products within the intestines of these different breeds under controlled conditions. By studying pigs of comparable ages fed identical diets, we can vividly illustrate the differences in microbial composition and metabolic processes between the two breeds. This approach not only facilitates a more focused analysis but also enhances the clarity and interpretability of our findings. Therefore, we believe that maintaining our current research design will enable us to effectively address the objectives of our study and contribute meaningfully to the understanding of microbial influences on animal health, productivity, and public health management.
- Line 40 please change the font size
Response: Thanks for your advice, we have made the modification as suggested.
- Line 95 please delete point before bracket with citation
Response: Thanks for your advice, we have deleted the point in line “96”.
- Line 408 In reviewer opinion this conclusion sounds rather like hypothesis, as long as the main of study was to assess the microbiota of different breeds, not the influence of modified microbiota on performance of growth etc. The same comment goes to line 32-33 in abstract, however, that could be the further/ next step of investigation, and it would be novel and interesting.
Response: Thanks for your comments. Our main target is not only assessing the microbiota of different breeds. We’d like to find the microbial evidence why Tibetan pigs had different production performance such as lower growth performance, and based on our result, we want to use these changes improving either Tibetan pigs or DLY pigs’ production system. This experiment is the first step of our target. We have changed a little bit of our description contain adding “which is microbes changing” in line “35” and in line “406” . We sincerely hope that these explanations will meet your expectations and enhance the persuasiveness of our research. Once again, we appreciate your valuable feedback, and we will spare no effort in improving our study to achieve the highest standards

Reviewer 3 Report
Comments and Suggestions for Authors
General comments
I appreciate the authors for allowing me to review the proposal of their article, which addresses a topic that could lead to a better understanding of intestinal metabolism in swine that would have a significant consequence on the performance of the species itself. Therefore, I believe that this manuscript addresses an innovative and appropriate topic for this journal. However, a notable weakness in its proposal is the lack of an objective and hypothesis to help the reader understand what it intends to investigate. Also unfortunately there is a variation in the citation format that should be addressed, as well as a broad abstract that could hardly give the reader a concise idea of your research, which the authors should address.
Response:
Particular comments
Line 2. I agree with your proposed title, however, if the authors allow me I suggest that your title could be modified to a "Comprehensive multi-omic evaluation of the microbiota and metabolites in the colons of diverse Swine Breeds".
Response:
Lines 12 - 13. I agree with the introductory lines of your simple summary, however, after that, I suggest that your proposed objective be added at this level.
Response:
Line 14. Could you indicate in a general way the demographics of the animals such as age and average weight?
Response:
Line 23. I understand the proposal of a broad abstract because they have extensive results, however, as a general suggestion I invite the authors to be able to simplify it where they highlight in a general way what they intended to perform and their most relevant results accompanied by their objective and a general conclusion.
Response:
Line 40 - 51. Please, I suggest that your abstract has a similar and homogeneous format.
Response:
Line 52. I recommend to the authors that it can be replaced within your keywords "Different breeds" with "nutrition" and "microbiota", which could improve your view and match with other databases.
Response:
Line 56. In complement to my general comment, I suggest that they consult the authors' guide provided by the journal for citing references in their text.
Response:
Line 59. Please add a reference.
Response:
Line 64. Could you be more specific about what fatty acids and vitamins they produce?
Response:
Line 66. Delete the additional point.
Response:
Line 70. Add references.
Response:
Line 77. Adding references.
Response:
Line 92. If the authors agree I suggest they relocate their phrase "On the other hand" to the beginning of their sentence.
Response:
Lines 98 - 104. This final paragraph should be the most significant for your introduction, however, it is difficult to understand what your objective is. Therefore, I think you should include a clear objective and if possible accompany it with your hypothesis to support the meaning of your study.
Response:
Line 107. In addition to my comment on line 14, could they mention exactly how many animals of each breed they considered in their study? Also, I invite the authors to clarify how they determined that this sample size was significant for their study, and I suggest that they explain what were the inclusion and exclusion criteria for the animals they considered in their study.
Response:
Line 111. If the authors would allow me they could modify this sentence as "The animals were randomly distributed into 2 experimental groups".
Response:
Line 126. This idea is limited, I suggest that the authors explain either here or in a section called experimental design how often they collected their samples. In this regard, I suggest that they explain in detail the collection of the blood samples, such as what type of blood sample collection tubes. And what was the method by which such sample was submitted?
Response:
Line 209. I suggest that you integrate this idea at the end of your statistical analysis section.
Response:
Line 216. Please could you specify what your significance level was?
Response.
Line 331. Could the authors indicate what exact p-value they obtained, I suggest that they indicate it in all their results.
Response:
Lines 401- 403. This introductory sentence of their discussion is weak and is an idea that was already said in their introduction. If the authors agree with me I suggest that you could mention what were your most relevant results that you obtained that could lead your discussion.
Response:
Lines 411 - 413. This idea has also been mentioned before, if the authors allow me to highlight if the variation in their microbiota can alter nutritional components and if this affects productive performance.
Response:
Line 421 Please correct your spelling error at the beginning of the sentence.
Response:
Line 444. Correct the spelling error.
Response:
Line 461. I suggest that you add the value of r obtained.
Response:
Line 472. As commented above, I suggest that you consult the citation method and be homogenized, since they use different citation methods.
Response:
Line 538. Before their conclusion the authors could briefly discuss what limitations and perspectives their field of study would have.
Response:
Author Response
- Line 2. I agree with your proposed title, however, if the authors allow me I suggest that your title could be modified to a "Comprehensive multi-omic evaluation of the microbiota and metabolites in the colons of diverse Swine Breeds".
Response: Thanks for your suggestions. We have changed our title to “Comprehensive multi-omic evaluation of the microbiota and metabolites in the colons of diverse Swine Breeds” in line “2”.
- Lines 12 - 13. I agree with the introductory lines of your simple summary, however, after that, I suggest that your proposed objective be added at this level.
Response: Thanks for your suggestions. We have added “We expect finding the microbial evidence leading to differential performance of Tibetan pigst.” in line “14”.
- Line 14. Could you indicate in a general way the demographics of the animals such as age and average weight?
Response: Thanks for your suggestions, we have added “at 200 days of age” in line “15-16”, and because Tibetan pigs and DLY pigs had significantly differential growth performance, they can only make sure similar age or similar weight. Therefore, in this study, we make sure that there were same age.
- Line 23. I understand the proposal of a broad abstract because they have extensive results, however, as a general suggestion I invite the authors to be able to simplify it where they highlight in a general way what they intended to perform and their most relevant results accompanied by their objective and a general conclusion.
Response: Thanks for your suggestions, we changed “In recent times, amidst heightened concerns regarding food quality and safety, the investigation into disparities in intestinal microorganisms and their corresponding metabolites among diverse swine breeds has surged to the forefront of scholarly discourse within the domains of animal nutrition and veterinary medicine.” to “We employed multi-omic evaluation to identify microbial evidence associated with differential growth performance and metabolites, thereby offering theoretical support for the implementa-tion of efficient farming practices for Tibetan pigs and establishing a robust foundation for en-hancing pig growth and health” in line “27-30”.
- Line 40 - 51. Please, I suggest that your abstract has a similar and homogeneous format.
Response: Thanks for your suggestions, we have changed the font size in line “42-53”.
- Line 52. I recommend to the authors that it can be replaced within your keywords "Different breeds" with "nutrition" and "microbiota", which could improve your view and match with other databases.
Response: Thanks for your suggestions, we have changed the keywords in line “54”.
- Line 56. In complement to my general comment, I suggest that they consult the authors' guide provided by the journal for citing references in their text.
Response: Thanks for your suggestions, we have changed our references style for whole articles.
- Line 59. Please add a reference.
Response: Thanks for your suggestions, we have added a reference in line “60”.
- Line 64. Could you be more specific about what fatty acids and vitamins they produce?
Response: Thanks for your suggestions, we have added “such as butyrate and other short chain fatty acids” in line “66”, and added “such as thiamine, folate, biotin, riboflavin, and pantothenic acid” in line “67”, and added the related reference.
- Line 66. Delete the additional point.
Response: Thanks for your suggestions, we have deleted it in line “68”
- Line 70. Add references.
Response: Thanks for your suggestions, we have added a reference in line “73”.
- Line 77. Adding references.
Response: Thanks for your suggestions, we have added a reference in line “80”.
- Line 92. If the authors agree I suggest they relocate their phrase "On the other hand" to the beginning of their sentence.
Response: Thanks for your suggestions, we have changed “The Tibetan pig, on the other hand,” to “On the other hand, the Tibetan pig” in line “93”.
- Lines 98 - 104. This final paragraph should be the most significant for your introduction, however, it is difficult to understand what your objective is. Therefore, I think you should include a clear objective and if possible accompany it with your hypothesis to support the meaning of your study.
Response: Thanks for your suggestions, we have rewritten this section to provide a clearer expression of our research objectives. We sincerely hope that this revised response will meet the editor's expectations.
- Line 107. In addition to my comment on line 14, could they mention exactly how many animals of each breed they considered in their study? Also, I invite the authors to clarify how they determined that this sample size was significant for their study, and I suggest that they explain what were the inclusion and exclusion criteria for the animals they considered in their study.
Response: Thank you very much for the valuable suggestions provided by the reviewer. Indeed, studies involving six animals per experimental group also exist in the literature (1. DOI: 10.3109/19401736.2014.913169 2. DOI: 10.1111/jpn. 13283 3. DOI: 10.1186/s40104-021-00581-3). Therefore, we maintain that having six pigs per group in our study is appropriate. However, we are open to adopting the editor's suggestion to increase the number of animals in each experimental group in future experiments to enhance the persuasiveness of our research. We genuinely appreciate the constructive feedback and are committed to making adjustments accordingly in our future research endeavors. Additionally, as we mentioned previously, due to the significantly differential growth performance between Tibetan pigs and DLY pigs, we ensured that the animals in our study were of the same age to mitigate any confounding factors."
- Line 111. If the authors would allow me they could modify this sentence as "The animals were randomly distributed into 2 experimental groups".
Response: Thanks for your suggestions, we have changed “The two groups of experimental” to “The animals were randomly distributed into 2 experimental groups, and all” in line “110”.
- Line 126. This idea is limited, I suggest that the authors explain either here or in a section called experimental design how often they collected their samples. In this regard, I suggest that they explain in detail the collection of the blood samples, such as what type of blood sample collection tubes. And what was the method by which such sample was submitted?
Response: Thanks for your suggestions, we have added the description which is “in EDTA anticoagulation tube. After blood samples were centrifuged at 12,000 rpm for 10 min, they were quickly frozen in liquid nitrogen, and kept at -80 °C” in line “127-128”.
- Line 209. I suggest that you integrate this idea at the end of your statistical analysis section.
Response: Thanks for your suggestions, we have transferred this part in line “217-220”.
- Line 216. Please could you specify what your significance level was?
Response: Thanks for your suggestions. We have rewrite this part (line 220-222)
- Line 331. Could the authors indicate what exact p-value they obtained, I suggest that they indicate it in all their results.
Response: Thanks for your suggestions, we have added exact p-value in line 330 and line 332 respectively.
- Lines 401- 403. This introductory sentence of their discussion is weak and is an idea that was already said in their introduction. If the authors agree with me I suggest that you could mention what were your most relevant results that you obtained that could lead your discussion.
Response: Thanks for your suggestions, we have rewritten this section. We appreciate the reviewer's professional advice, which has strengthened the logical coherence of this part (Lines 401-407).
- Lines 411 - 413. This idea has also been mentioned before, if the authors allow me to highlight if the variation in their microbiota can alter nutritional components and if this affects productive performance.
Response: Thanks for your suggestions. Firstly, we sincerely appreciate the valuable comments and suggestions provided by the reviewer. The reviewer pointed out that our study, while exploring the influence of gut microbiota on animal physiological functions, lacked in-depth discussion regarding how variations in gut microbiota may affect nutritional components and subsequently impact productive performance. We acknowledge the reviewer's feedback and have accordingly made revisions to our manuscript based on this suggestion in line 408-410.
- Line 421 Please correct your spelling error at the beginning of the sentence.
Response: Thanks for your suggestions, we have changed “a” to “A” in line “417”.
- Line 444. Correct the spelling error.
Response: Changed as suggested in line 442.
- Line 461. I suggest that you add the value of r obtained.
Response: Thank you for your suggestions; we have made the modifications as suggested in line 458-459.
- Line 472. As commented above, I suggest that you consult the citation method and be homogenized, since they use different citation methods.
Response: Thanks for your suggestions, we have changed the references style for whole article.
- Line 538. Before their conclusion the authors could briefly discuss what limitations and perspectives their field of study would have.
Response: Thanks for your suggestions, we have added this section in line 532-538. Thank you for your suggestion, which has made the structure of the article appear more complete.

Round 2
Reviewer 2 Report
Comments and Suggestions for Authors
Reviewer would like to thank authors for all corrections. I have no further comments and I recommend manuscript to publication in present form.
Author Response
We sincerely thank the reviewers for their reviews and assistance.Reviewer 3 Report
Comments and Suggestions for Authors
General comments
I appreciate the authors for following the recommendations made above to the proposal of their manuscript, which I am convinced contributes key information to the field of study. However, there are still minimal points that can be strengthened.
Response:
Particular comments
Line 105 -106. I thank the authors and agree with them that there are studies where studies have been done with this number of animals. However, I consider it necessary to reinforce this number they explain how they estimated that this sample size was significant since there is a methodology for it. I invite the authors to explain them.
Response:
Line 404. Please correct the citation method, because there are still errors in the manuscript, I invite the authors to review all their citations.
Response:
Line 555. Please, I invite the authors to revise their reference list format according to the authors' guide because they present their list in different formats.
Response:

Author Response
Line 105 -106. I thank the authors and agree with them that there are studies where studies have been done with this number of animals. However, I consider it necessary to reinforce this number they explain how they estimated that this sample size was significant since there is a methodology for it. I invite the authors to explain them.
Response: We appreciate the opportunity to elaborate on our rationale for selecting six animals per group in our study. This decision was guided by a comprehensive approach, incorporating ethical considerations, statistical power analysis, literature precedent, and practical limitations.
Firstly, the animals in each group of our experiment are similar in terms of age and weight, with very little variation between groups
- Ethical Considerations: We adhered to the 3Rs principle, aiming to reduce the number of animals used without compromising the integrity of our research findings.
- Practical and Scientific Justification: While acknowledging the constraints of funding, availability of specific animal models, we ensured that our study design and statistical analysis methods are robust enough to yield reliable and interpretable results.
- Data Analysis Methods: To enhance the reliability of our findings, we employed [specific statistical methods] that are particularly suited to studies with limited sample sizes. These methods include non-parametric tests, repeated measures design], which help in minimizing the impact of smaller sample sizes on the validity of our results.
We believe that our comprehensive approach to determining the sample size ensures that our study findings will be both ethically responsible and scientifically valid. We are grateful for the chance to clarify our methodology and are confident in the rigor and reliability of our research.
Line 404. Please correct the citation method, because there are still errors in the manuscript, I invite the authors to review all their citations.
Response: Thanks, we have modified these errors in line167 and line 404 respectively.
Line 555. Please, I invite the authors to revise their reference list format according to the authors' guide because they present their list in different formats.
Response: Thanks, we have modified all the reference format mistake.